# MINIF2F: A CROSS-SYSTEM BENCHMARK FOR FORMAL OLYMPIAD-LEVEL MATHEMATICS

**Kunhao Zheng**
École Polytechnique
`kunhao.zheng@polytechnique.edu`

**Jesse Michael Han**
OpenAI
University of Pittsburgh
`jessemichaelhan@openai.com`

**Stanislas Polu**
OpenAI
`spolu@openai.com`

## ABSTRACT

We present miniF2F, a dataset of formal Olympiad-level mathematics problems statements intended to provide a unified cross-system benchmark for neural theorem proving. The miniF2F benchmark currently targets Metamath, Lean, Isabelle (partially) and HOL Light (partially) and consists of 488 problem statements drawn from the AIME, AMC, and the International Mathematical Olympiad (IMO), as well as material from high-school and undergraduate mathematics courses. We report baseline results using GPT-$f$ (Polu & Sutskever, 2020), a neural theorem prover based on GPT-3 (Brown et al., 2020) and provide an analysis of its performance. We intend for miniF2F to be a community-driven effort and hope that our benchmark will help spur advances in neural theorem proving.

## 1 INTRODUCTION

Shared benchmarks and datasets have historically played a crucial role in driving advances in large-scale applications of deep learning, e.g. in computer vision (Deng et al., 2009) and natural language processing (Wang et al., 2019; Rajpurkar et al., 2016; Paperno et al., 2016). *Neural theorem proving* is a rapidly developing area which aims to apply techniques from deep learning to interactive theorem proving. To date, most contributions in this area have focused on individual theorem proving systems, each with a separately-implemented mathematics library and with results reported on a dataset-specific test split; examples include the HOList (Bansal et al., 2019a), CoqGym (Yang & Deng, 2019) and LeanStep (Han et al., 2021) theorem proving environments and benchmarks. However, benchmarks from this paradigm are not ideal for measuring the mathematical reasoning ability of neural theorem provers for several reasons. Library-specific train/test splits are siloed by construction, dependent on how theorems and lemmas are split in these libraries, and as such are not directly comparable across systems. Moreover, formal mathematics libraries are closer to software repositories than informal mathematical exposition, and many lemmas are implementation-specific artifacts without precise informal mathematical or cross-system translations.

To date, the neural theorem proving community has not organized its efforts around a cross-system benchmark. To address this need and to provide a common resource to research groups working on formal theorem proving, we present miniF2F, a unified cross-system benchmark of formal mathematics of progressively increasing difficulty, centering around Olympiad-level problem statements (AMC, AIME, IMO) as well as high-school and undergraduate maths classes. Both the content and name of miniF2F are inspired by the IMO Grand Challenge (Selsam et al., 2019): to build an AI that can win a gold medal in the International Mathematical Olympiad in a *formal-to-formal* (F2F) format. More precisely, the agent must receive IMO problems written in a formal mathematical format, and must produce a formal (i.e. machine-checkable) proof for that problem.

We intend for miniF2F to serve as a stepping stone for different formal systems towards the IMO Grand Challenge (Selsam et al., 2019), as it is end-to-end verifiable, cross-platform and spans a wide range of difficulty. While we report baseline results on miniF2F using GPT-$f$, a language model

based on GPT-3 which has been finetuned for theorem proving, language models are not a mandatory approach for Olympiad problems and this assumption is not reflected in miniF2F, preserving the generality and widespread applicability of the benchmark to systems similar to DeepHOL (Bansal et al., 2019a) or Holophrasm (Whalen, 2016).

## 2    BACKGROUND AND RELATED WORK

### BENCHMARKS

In the closely related field of (first-order) *automated theorem proving* (ATP), the TPTP (Sutcliffe, 2017) benchmark is a library of test problems in a unified format for ATP systems. In interactive theorem proving, the "Freek 100" (Wiedijk, 2008) tracks progress across various interactive theorem provers on a list of 100 mathematical theorems. Wu et al. (2021) built a simplified formal proof environment INT with an associated synthetic inequality benchmark. Competitions and communal challenges have also spurred development in formal theorem proving. The CADE ATP System Competition (CASC) (Sutcliffe, 2016) is a competition that evaluates the performance of first-order automated theorem proving systems. Proof Ground (Haslbeck et al., 2019), part of the ITP conference, is an interactive proving contest (for humans) that supports Coq, Isabelle, and Lean, which focuses on evaluating the formalization effort of proof to given problems within limited time. Finally, the IMO Grand Challenge (Selsam et al., 2019), a proposal from researchers working on the interactive proof assistant Lean, aims to build a system capable of solving IMO problems in the formal-to-formal format.

Due to its convenient framing as a natural language processing task, the domain of informal mathematical reasoning has received more attention than the formal one. MATH (Hendrycks et al., 2021) is a mathematics benchmark comprising 12,500 statements in natural language where exercises are classified into 5 levels of difficulty across various domains. Each exercise is combined with a detailed step-by-step proof in natural language. Scaling state-of-the-art models shows little amelioration on MATH, which requires advanced mathematical reasoning capabilities. miniF2F includes a number of formalized statements from MATH. NaturalProofs (Welleck et al., 2021) is another benchmark of natural proof in mathematics , containing 32k theorem statements and proofs. It essentially contains the proofs in ProofWiki and other resources. While MATH is more oriented towards mathematics exercises, NaturalProofs is focused on proofs of general mathematics theorems. Saxton et al. (2019) built a mathematics dataset with $2 \times 10^6$ training data and $10^4$ test data, presented in a question-answering format where each statement is paired with a question written in natural language and a direct answer without proof.

### NEURAL THEOREM PROVING

HOList (Bansal et al., 2019a;b; Paliwal et al., 2020) provides an environment as well as a benchmark for HOL Light. They also proposes various deep reinforcement learning approaches for theorem proving and report a pass rate of $59.91\%$ on their benchmark. Yang & Deng (2019) built CoqGym, a large-scale dataset, which comes also with a learning environment, of 71k human-written proofs in Coq proof assistant. They report a $30.0\%$ pass rate on the held-out test theorems in CoqGym. Polu & Sutskever (2020) applied a decoder-only transformer similar to GPT-3 (Brown et al., 2020) to proof steps prediction in Metamath combined with a log-probability based proof search. They also proposed a methodology to train a value function to further guide proof search, achieving a $56.22\%$ pass rate on the held-out test set. Large language models were applied to Lean by Han et al. (2021). They created an environment around the Lean prover targeted to machine learning and propose a dataset extracted from low level proof artifacts that is shown to boost performance when used as a self-supervised co-training objective. They report a $48.4\%$ pass rate on held-out test statements from `mathlib`, Lean's mathematical library (mathlib Community, 2020).

## 3    MINIF2F BENCHMARK

miniF2F is a dataset of manually formalized statements of Olympiad type problems, aligned in Lean, Metamath, and Isabelle (partial at the time of writing), providing a cross-platform benchmark for formal mathematical reasoning. Olympiad type problems are of particular interest to compare

Table 1: Number of statements and their provenance in miniF2F v1

| | | | Test Set | Validation Set |
|---|---|---|---|---|
| TOTAL | | | 244 | 244 |
| **IMO** | | | 20 | 20 |
| **AIME** | | | 15 | 15 |
| **AMC** | | | 45 | 45 |
| **MATH** | Algebra | Level 5 | 14 | 14 |
| | | Level 4 | 14 | 14 |
| | | Level 3 | 14 | 14 |
| | | Level 2 | 14 | 14 |
| | | Level 1 | 14 | 14 |
| | Number Theory | Level 5 | 16 | 16 |
| | | Level 4 | 11 | 11 |
| | | Level 3 | 11 | 11 |
| | | Level 2 | 11 | 11 |
| | | Level 1 | 11 | 11 |
| **CUSTOM** | Algebra | | 18 | 18 |
| | Number Theory | | 8 | 8 |
| | Induction | | 8 | 8 |

automated provers across different formal systems as the theories required to solve them are well identified and they generally do not require the definition of new mathematical concepts (a capability that remains beyond the current neural theorem proving state of the art).

The formalized statements in miniF2F are drawn from multiple sources, ranging from high school and undergraduate level exercises to Olympiad problems. miniF2F also covers different sub-subjects in mathematics as well as proof strategies, focusing on the types of exercises whose statements are expressible in most formal systems. This leads to a systemic focus on algebra, number theory and inequalities because, for example, geometry and combinatorial problems are generally challenging to formalize due to only nascent efforts in these areas in most formal systems. The statements in miniF2F are all manually formalized and selected to cover a variety of difficulty levels for both humans and machines. Formal proofs for these statements are optionally attached.

miniF2F draws from AIME, AMC, IMO problems as well as problems from the MATH (Hendrycks et al., 2021) informal dataset. Formalizing problems from the MATH dataset serves two purposes. First, problems in MATH are segmented by difficulty level (from 1 to 5), randomly selecting a subset from each of these difficulty levels allows miniF2F to cover a wider range of difficulty. Second, it provides the community an opportunity to compare capabilities of formal automated prover to their informal counter-parts as discussed in later sections.

miniF2F comprises a test set and a validation set, which are a stratified random split from the statements we formalized such that each set equally covers each problem type and difficulty (when available). Table 1 shows a detailed distribution of these statements.

**Versioning** miniF2F is an evolving effort and new statements will continuously be added. Periodically, we will freeze versions of the benchmark. The current version of the benchmark is v1[1] and results in this paper are reported using this version. v1 comprises 244 test and 244 valid statements. The set of statements of each version is guaranteed to remain stable, only allowing fixes in case errors are later discovered.

**Rules of engagement and License** miniF2F is meant to serve as a shared resource for research groups working on applying deep learning to formal theorem proving. There is no formal process to submit evaluation results and researchers are simply invited to cite miniF2F indicating the version used in their evaluations. We also encourage them to contribute proofs found by their approaches back to the benchmark. The parts of the benchmark associated with each theorem prover (Metamath,

---

[1]https://github.com/openai/miniF2F/tree/v1

Lean, Isabelle) are meant to be licensed in a way that is aligned with the licensing usage associated with the theorem prover's main library. As a result, the Metamath version of the benchmark is released under the MIT License, while the Lean and Isabelle versions are released under the Apache License.

**Formalization effort and challenges**   We found that, for trained practitioners (but not necessarily experts, including students recently introduced to formal systems), formalizing a statement takes about 15 minutes on average, and reviewing a formalized statement, about half of that on average. Note that not all exercises are directly or naturally formalizable. In particular, multi-choice questions, word problems, and exercises that require to explicit a witness or a set as part of the answer present interesting challenges:

*multi-choice questions*[2] these problems are generally straightforwardly formalizable by reformulating the statement using the right answer only, and could be made "fair" in a competitive setup by formalizing all possible choices and running automated provers on all of them, attributing points only if a proof of the correct answer is provided.

*word problems*[3] where significant information is presented in natural language generally require non-trivial efforts to be formalized. We generally formalized them by explicitly modeling the mathematics concepts and expression presented in natural language while attempting as best as possible to preserve the mathematical difficulty of the original problem. Sometime the formalization work is most of the difficulty associated with the original question; in such cases we would discard the problem entirely.

*problems that require to explicit a set or witness*[4] (e.g. find all ... such that ...) are not directly formalizable. The best approximation we relied on for these was to formalize the statement with the witness or answer provided, turning such exercises into the generation of a proof that the answer is correct, and if needed, that it is the unique one–which is, at times, a much easier exercise. A non negligible portion of IMO problems are as such, which we foresee could become a challenge in the future, to fairly compare humans to automated proving systems in a competitive setup.

**Porting effort**   In addition to Metamath, Lean, Isabelle (work in progress) and HOL Light (work in progress), we are eager to extend the coverage of miniF2F to Coq, and will welcome any effort in that direction or to extend miniF2F to further systems.

## 4   EXPERIMENTS

In this section, in order to study baseline performances associated with existing systems, we report pass rates achieved by GPT-$f$ (Polu & Sutskever, 2020) applied to Metamath, GPT-$f$/PACT (Polu & Sutskever, 2020; Han et al., 2021) applied to Lean as well as a baseline prover implemented in Lean denoted as the `tidy` baseline. Pass rates are reported as Pass@$N$ where $N$ is the number of proof search attempts per statement. Pass@$N$ is computed by running more attempts per statement, averaged to get an unbiased, low-variance estimate.

### 4.1   METAMATH

Metamath is powered by a meta logic system based on a single substitution rule. It's characterized by its simplicity which makes it convenient to study machine learning. Proofs in Metamath are, as a consequence of the low-level proofsteps, much longer than in other systems as there is no assistance from high-level tactics. Proofs which are trivial in other systems (e.g. n-digit addition or simple ring arithmetic transformations) can be quite tedious in Metamath. The absence of tactics is both

---

[2]Example: `amc12a_2020_p10` in `https://github.com/openai/miniF2F/blob/main/lean/src/test.lean`

[3]Example: `mathd_algebra_398` in `https://github.com/openai/miniF2F/blob/main/lean/src/test.lean`

[4]Example: `imo_1997_p5` in `https://github.com/openai/miniF2F/blob/main/lean/src/test.lean`

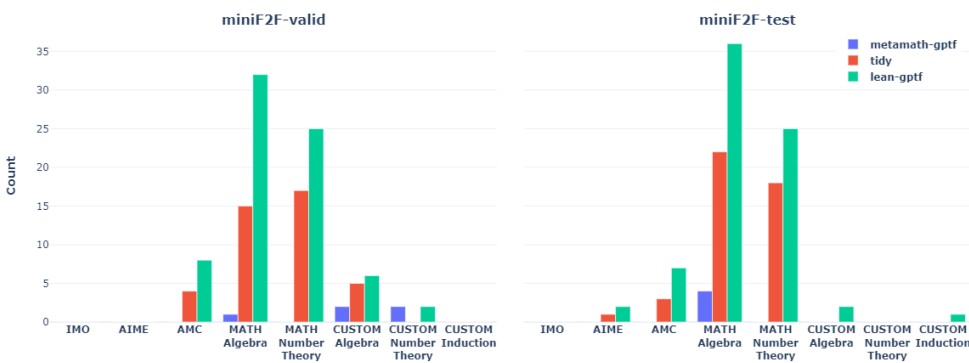

Figure 1: Counts of successfully proved statements in miniF2F. Green bar: results from Lean GPT-f. Red bar: best result from the `tidy` baseline. Blue bar: results from Metamath GPT-f.

a benefit, as the models sees and learns on everything, and a challenge, as proofs of even simple exercises require hundreds of proofsteps.

### 4.1.1 GPT-F

We report the pass rate of GPT-$f$ applied to Metamath as described in Polu & Sutskever (2020). We use a model with 700m learnable parameters. The model is trained on an updated dump of the set.mm library (but similar synthetic datasets), using the log-probability based search as reported in Table 8 of the GPT-$f$ paper (Polu & Sutskever, 2020).

The model achieves a Pass@1 of $1.3\%$ and a Pass@8 of $1.6\%$ on miniF2F-test. As expected, these numbers are quite low due to the length of typical proofs for even simple math exercises. The average proof length is also reported in Table 3.

### 4.2 LEAN

In comparison to Metamath, Lean benefits from a large number of powerful tactics to assist formalization efforts. Typical Lean proofs are much shorter than Metamath's. This is also a formal system of interest as it has received a lot of attention from the mathematical community as recent theories have successfully been formalized in Lean (Perfectoid Spaces (Buzzard et al., 2019), Liquid Tensor experiment (Scholze, 2020)).

Lean is also associated with the IMO Grand Challenge (Selsam et al., 2019) which aims to organize a formal-to-formal challenge during the upcoming IMO competitions.

### 4.2.1 TIDY BASELINE

We use the generic best-first search algorithm presented in PACT (Han et al., 2021). The algorithm works as follows: Given a list of tactics $L$ with priority, we maintain a priority queue $Q$ of tactic states whose priority is given by the priority of the last applied tactic in $L$ that led to it. While $Q$ is not empty, we pop the top tactic state $t$ from $Q$. We iterate through $L$ and apply each tactic to $t$. If no error is raised, we capture the returned tactic states from Lean and insert them back into $Q$.

We use the same terminology as in PACT (Han et al., 2021): maximum queue size $\omega_{max}$, depth limit $d_{max}$. We also enforce a budget of $i_{max}$ iterations of the outer loop. When $Q$'s size reach $q_{max}$, all the tactic states to be inserted are discarded. We do not expand the next tactic state when the depth is beyond $d_{max}$. This loop is run until a proof is found or the iterations budget is exhausted.

For consistency checking, we run the `tidy` baseline under the same settings and on the same test set as in PACT (Han et al., 2021) except that we don't set a global timeout. Our implementation

achieved a 10.5% pass rate on mathlib's test split. This result is comparable to the reported 9.9% in PACT given the waived global timeout.

In addition to the curated list of tactics $L$ used in PACT (Han et al., 2021), we added 4 high-level tactics $HL =$[nlinarith, linarith, ring_nf, norm_num] to $L$ with higher priorities than the others. We report our pass rate on miniF2F in Table 2.

Table 2: The table shows the number of solved statement in miniF2F when running the tidy baseline with different values of $i_{max}$ as well Lean's built-in tidy tactic. All tidy baseline experiments are run with $\omega_{max} = 128$, $d_{max} = 8$ using $L + HL$. Despite the tidy baseline being deterministic, it is still subject to per-tactic application timeouts, explaining the number 43 reported on miniF2F-test for $i_{max} = 32$.

| parameters | miniF2F-valid | miniF2F-test |
|---|---|---|
| Lean's tidy tactic | 12 / 244 | 13 / 244 |
| $i_{max} = 1$ | 21 / 244 | 23 / 244 |
| $i_{max} = 2$ | 31 / 244 | 29 / 244 |
| $i_{max} = 4$ | 38 / 244 | 41 / 244 |
| $i_{max} = 8$ | 41 / 244 | 44 / 244 |
| $i_{max} = 16$ | 41 / 244 | 44 / 244 |
| $i_{max} = 32$ | 41 / 244 | 43 / 244 |
| $i_{max} = 64$ | 41 / 244 | 44 / 244 |
| $i_{max} = 128$ | 41 / 244 | 44 / 244 |

### 4.2.2 GPT-F/PACT

We report the pass rate of GPT-$f$/PACT as described in Han et al. (2021). We use a model with 700M learnable parameters. The model is trained on an updated dump[5][6] of the mathlib library using the PACT methodology denoted in the paper as mix2 > mix1 + tactic in Figure 6.

The model achieves a Pass@1 of $24.6\%$ and a Pass@8 of $29.2\%$ on miniF2F-test. The average proof length is also reported in Table 3.

Table 3: Baseline performance on Metamath and Lean. All proof searches are provided with a 128 expansions budget. GPT-$f$ attempts $e = 16$ tactics per expansion while the tidy baseline attempts $e = 17$ tactics per expansion ($L + HL$, see section 4.2.1). Reported proof lengths are averages over all the proofs found in each run. Note that the tidy baseline being deterministic, there is no point attempting a proof search more than once.

| Formal System | Model | miniF2F-valid | | | miniF2F-test | | |
|---|---|---|---|---|---|---|---|
| | | Proof Length | Pass@1 | Pass@8 | Proof Length | Pass@1 | Pass@8 |
| Metamath | GPT-$f$ | 16.2 | 1.0% | 2.0% | 20.3 | 1.3% | 1.6% |
| Lean | tidy | 1.7 | 16.8% | - | 1.8 | 18.0% | - |
| Lean | GPT-$f$ | 2.6 | 23.9% | 29.3% | 2.5 | 24.6% | 29.2% |

## 4.3 DISCUSSION

### 4.3.1 ACCESS TO HIGH-LEVEL TACTICS

One goal of miniF2F is to study the comparison of performance across formal systems. In this section we reported the performance of the same methodology (GPT-$f$ (Polu & Sutskever, 2020))

---

[5]https://github.com/jasonrute/lean_proof_recording/commit/
8499f10c2e10dd533152070ed933c4f0b21ecdc0
[6]https://github.com/jesse-michael-han/lean-step-public/commit/
a2b83c237bfe4d6f1c48bb48bc0769b5940e614a

applied to both Lean and Metamath. Both models are pre-trained on WebMath (Polu & Sutskever, 2020) and respectively trained on datasets extracted from Lean (Han et al., 2021) and Metamath (Polu & Sutskever, 2020). The overall compute deployed at training is comparable in both setup and exactly equivalent at test-time, yet the achieved performance appears drastically superior when applied to Lean. We hypothesize that this is mainly explained by the model's access to high-level tactics when applied to Lean, enabling the model to learn how to guide Lean's automation in an effective way.

An example of this high-level guidance behavior is well exemplified by the following proof of the statement `algebra_sqineq_2unitcircatblt1` where the model heavily relies on Lean's `nlinarith` solver but provides it with essential premises to successfully guide the search.

```
theorem algebra_sqineq_2unitcircatblt1
  (a b : ℝ)
  (h₀ : a^2 + b^2 = 2) :
  a * b ≤ 1 :=
begin
  nlinarith [sq_nonneg a,sq_nonneg b,sq_nonneg (a - b)]
end
```

(The statement above (`algebra_sqineq_2unitcircatblt1`) requires to prove the assertion $\forall a, b \in \mathbb{R}, a^2 + b^2 = 2 \rightarrow a \cdot b \leq 1$).

In Metamath, GPT-$f$ fails to find a proof as it requires a very large number of steps to appropriately rewrite the goal in a way that is amenable to the use of set.mm's existing theorems. The `tidy` baseline also fails to find a proof of that statement as `nlinarith` is not capable of solving the goal without being passed extraneous premises.

These results motivate the use of neural theorem proving with formal systems that expose powerful high level tactics and also suggest the potential of a closer collaboration between formal systems and machine learning practitioners. It also motivates the use of generative models in that setup as the arguments required by high-level tactics to succeed on non trivial problems generally do not exist in the context of the statement and therefore have to be generated ex-nihilo.

### 4.3.2 COMPARISON OF INFORMAL AND FORMAL SETUPS

The use of formal systems for neural theorem proving is often motivated by the role of the formal system as a verifier, enabling more advanced neural search strategies than possible in a fully informal setup where the generation of a model can't be verified automatically, as well as the access to powerful tactics. Our formalization of a subset of the MATH (Hendrycks et al., 2021) informal dataset provides an interesting approximate quantification of the benefit of having access to a formal system in the context of neural theorem proving. Approximate, because we only formalized a small subset of the MATH statements, but nonetheless useful since we drew uniformly from the 5 difficulty levels.

In Hendrycks et al. (2021), the performance of GPT-3 (which is a larger model than the GPT-f model studied here) is reported to be 6.0% in the algebra category and 3.9% in the number theory category. GPT-$f$ applied to Lean by comparison achieves 51.4% in the algebra category and 41.7% in the number theory category. It is also worthwhile to note that the `tidy` baseline also highly outperforms (31.4% in algebra and 30.0% in number theory) GPT-3 in an informal setup demonstrating the benefit of proof automation alone.

### 4.3.3 LIMITATION

With miniF2F being cross-system as the goal, types of problems that are less expressible in certain systems such as geometry and combinatorial problems are less covered. The shift of distribution of problem types may result in skewing the research direction of models when benchmarking on miniF2F. Directionally we aim to fix it and extend the coverage of miniF2F as we grow the benchmark. However, works and efforts on the corresponding library of other systems are required as well.

## 5 CONCLUSION

We presented miniF2F, a dataset of formal Olympiad-level mathematics problem statements, meant to serve as an initial effort towards cross-system benchmarking of neural mathematical reasoning capabilities in formal environments. We reported the performance of the neural theorem prover GPT-$f$ (Polu & Sutskever, 2020) on both the Lean and Metamath parts of miniF2F as well as the performance of our non-neural `tidy` baseline applied to Lean. Then, we discussed these baselines and put them in perspective with previously reported comparable results in informal environments (Hendrycks et al., 2021).

Finally, we hope that miniF2F will prove to be useful to the scientific community working on neural theorem proving and spur advances in this domain.

ACKNOWLEDGMENTS

We are grateful to Wenda Li and Xavier Martinet for contributing the Isabelle and HOL Light statements currently available in miniF2F, paving the way towards a full support of Isabelle and HOL Light, as well as their feedback and encouragement in the process. We thank Harri Edwards for his comments that greatly improved the manuscript.

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

## A    EXAMPLE OF STATEMENT IN MINIF2F

Table 4: Problem 11 of 2000 AMC 12 is formalized with proof in different languages in miniF2F. The proof is optionally attached thus not part of the benchmark. The proof in Metamath is too long to be fully displayed.

| Natural Language | Two non-zero real numbers, $a$ and $b$, satisfy $ab = a - b$. Which of the following is a possible value of $\frac{a}{b} + \frac{b}{a} - ab$? (A) -2 (B) $\frac{-1}{2}$ (C) $\frac{1}{3}$ (D) $\frac{1}{2}$ (E) 2 |
|---|---|
| Metamath | ```
${
amc12-2000-p11.0 $e |- ( ph -> A e.   RR ) $.
amc12-2000-p11.1 $e |- ( ph -> B e.   RR ) $.
amc12-2000-p11.2 $e |- ( ph -> A =/= 0 ) $.
amc12-2000-p11.3 $e |- ( ph -> B =/= 0 ) $.
amc12-2000-p11.4 $e |- ( ph -> ( A x.   B ) =
  ( A - B ) ) $.
amc12-2000-p11 $p |- ( ph -> ( ( ( A / B ) +
  ( B / A ) ) - ( A x.   B ) ) = 2 )
$=
( cdiv co caddc cmul cmin c2 cexp eqcomd ...   $.
$}
``` |
| Lean | ```
theorem amc12_2000_p11
  (a b :   ℝ)
  (h₀ :   a ≠ 0 ∧ b ≠ 0)
  (h₁ :   a * b = a - b) :
  a / b + b / a - a * b = 2 :=
begin
  field_simp [h₀.1, h₀.2],
  simp only [h₁, mul_comm, mul_sub],
  ring,
end
``` |
| Isabelle | ```
theorem amc12_2000_p11:
  fixes a b::real
  assumes "a \<noteq> 0" "b \<noteq> 0"
    and "a * b = a - b"
    shows "a / b + b / a - a * b = 2"
  using assms
  by (smt (verit, ccfv_threshold)
    diff_divide_distrib
    div_self divide_divide_times_eq
    eq_divide_imp nonzero_mult_div_cancel_left)
end
``` |

# B   PERFORMANCE BY DIFFICULTY ON STATEMENTS FORMALIZED FROM MATH DATASET

The MATH dataset assigns a difficulty ranging from 1 to 5 to each of its problem. Tables 5 and 6 report the number of proved statement split by difficulty level on the algebra and number theory categories.

Table 5: Counts of successfully proved statements formalized from MATH-Algebra in miniF2F v1 split by difficulty. This table corresponds to "MATH Algebra" in Figure 1.

|  | miniF2F-valid | | | | | miniF2F-test | | | | |
| --- | --- | --- | --- | --- | --- | --- | --- | --- | --- | --- |
| Difficulty Level | 1 | 2 | 3 | 4 | 5 | 1 | 2 | 3 | 4 | 5 |
| Metamath/GPT-$f$ | 1 | 0 | 0 | 0 | 0 | 2 | 0 | 1 | 0 | 1 |
| Lean/tidy | 6 | 4 | 2 | 2 | 1 | 6 | 4 | 7 | 3 | 1 |
| Lean/GPT-$f$ | 9 | 7 | 8 | 6 | 2 | 8 | 7 | 10 | 7 | 3 |

Table 6: Counts of successfully proved statements formalized from MATH-Number theory in miniF2F v1 split by difficulty. This table corresponds to "MATH Number Theory" in Figure 1.

|  | miniF2F-valid | | | | | miniF2F-test | | | | |
| --- | --- | --- | --- | --- | --- | --- | --- | --- | --- | --- |
| Difficulty Level | 1 | 2 | 3 | 4 | 5 | 1 | 2 | 3 | 4 | 5 |
| Metamath/GPT-$f$ | 0 | 0 | 0 | 0 | 0 | 0 | 0 | 0 | 0 | 0 |
| Lean/tidy | 8 | 3 | 2 | 2 | 2 | 7 | 4 | 3 | 2 | 2 |
| Lean/GPT-$f$ | 9 | 5 | 5 | 4 | 2 | 10 | 5 | 5 | 3 | 2 |

More broadly, Lean GPT-$f$ is capable of solving any problem that the tidy baseline or Metamath GPT-$f$ can solve in MiniF2F. Qualitatively, the problems on which it fail either require multiple non-trivial reasoning steps (outside a few exceptions, problems requiring more than 2 non-trivial steps of mathematical reasoning are generally out of reach of these baselines) or require a cut introduction that is hard to generate, such as generating a non trivial witness.

