# OpenReview forum: "miniF2F: a cross-system benchmark for formal Olympiad-level mathematics"
_ICLR.cc/2022/Conference — ICLR 2022 Poster_

### Official Review · Reviewer_wdpj · 2021-10-19

**Correctness:** 4
**Technical Novelty And Significance:** 3
**Empirical Novelty And Significance:** 3
**Recommendation:** 8
**Confidence:** 4

**Main Review:**

$\textbf{strengths}$
- The paper formalizes a decent amount of cross-system Olympiad-level benchmark of 488 problems. Cross-system support on Metamath, Lean and Isabelle provides benefit on comparing automation and tactics of systems. Olympiad-level problems are also interesting to both researchers and general public.
- The inclusion of formalization of a subset of MATH benchmark also enables comparing provers in formal and informal format.
- The paper is well-written with good literature review on theorem proving benchmarks.

$\textbf{weaknesses}$
- The paper could justify more on what type of problems were selected in $\textbf{miniF2F}$. The benchmark mostly focuses on algebra, number theory and inequalities. Will this benchmark in some way skew the research direction of the community to only focus on developing algorithms particularly suitable for solving these types of problems that may or may not generalize well to other types of problems such as geometry problems?
- It would be interesting to see if the gap on pass rates between Metamath and Lean could be reduced when the models are trained or fine-tuned on a subset of $\textbf{miniF2F}$ in addition to pre-training. This could provide more evidence if the gap is mostly due to access to high-level tactics.

**Summary Of The Paper:**

The paper proposes a benchmark of formal Olympiad-level math problems focusing on algebra, number theory and inequalities with cross-system support on Metamath, Lean and Isabelle (in development). The paper evaluates performance of $\textit{GPT-f}$ on Metamath and Lean, and a custom baseline $\textit{tidy}$ on Lean as well.

**Summary Of The Review:**

The paper makes a solid step forward on creating cross-system Olympiad-level formal math benchmark and should have profound benefit on the community with its continual development.

---

> ### Author Response · Authors · 2021-11-20
> **Response to Reviewer wdpj**
>
> Thank you very much for your review and the time you dedicated to this.
>
> > The paper could justify more on what type of problems were selected in miniF2F. The benchmark mostly focuses on algebra, number theory and inequalities. Will this benchmark in some way skew the research direction of the community to only focus on developing algorithms particularly suitable for solving these types of problems that may or may not generalize well to other types of problems such as geometry problems?
>
> The cross-system nature of MiniF2F means that we are bounded by the expressivity of the maths library of the systems we target and plan to target, which indeed skews us at this stage toward algebra, number theory, or inequalities. That being said, it is interesting to note that olympiad problems are also generally skewed towards certain types of problems.
>
> Also, there is active work, as an example in Lean, towards enabling the formalization of other types of olympiad problems (probability, geometry, …). Lean, as an example, will probably soon reach a state where we can formalize 100% of problems from AMC/AIME; but this will obviously require non-trivial work in other systems as well.
>
> Indeed, that skew may induce a skew in the type of systems we build as a community. but we fundamentally believe that miniF2F, even if skewed, is better than nothing. Directionally the goal is obviously to fix this as we grow the benchmark
>
> > It would be interesting to see if the gap on pass rates between Metamath and Lean could be reduced when the models are trained or fine-tuned on a subset of miniF2F in addition to pre-training. This could provide more evidence if the gap is mostly due to access to high-level tactics.
>
> Thank you for your suggestion. We see the primary goal of the paper as presenting the benchmark more than studying in detail the systems we used to produce the baselines we propose, so we feel studying this might be out of the scope of the paper. However, we do have empirical evidence that expert iteration, while providing an uplift for both Lean GPT-f and Metamath GPT-f, does not sensibly reduce the gap between their performance.

---

### Official Review · Reviewer_Jr4E · 2021-11-02

**Correctness:** 4
**Technical Novelty And Significance:** 2
**Empirical Novelty And Significance:** 3
**Recommendation:** 5
**Confidence:** 3

**Main Review:**

Strengths:
- The advantages of this benchmark are that it is cross-system and that it covers a variety of mathematical topics at the Olympiad level.
- The motivation for the particular assemblage of mathematical topics is solid: miniF2F is intended as an intermediate step toward the IMO as an ATP task, which is out of reach for current systems. This is the first effort to unify various Olympiad topics in one dataset, and the problems cover a wide scope of tactics and difficulties.

Weaknesses / questions:
- The benchmark is not really as cross-system as claimed in the abstract. Only 12% of the training statements are available in Isabelle.
- How were the Olympiad and Custom problems chosen?
- The way in which multiple-choice problems are formalized gives additional information to the solver:
   - In Table 4, the AMC problem asks which value of an expression is *possible* (quantifier on a and b), but the formalization drops the quantifiers and asks to prove equality. This would be incorrect under minor changes in the conditions. It would seem more appropriate to formalize this with "x = -2 or x = 1/2 or ..." as a hypothesis.
   - Problem 22 of AMC 12B 2020 asked to find the maximum value of a certain function, out of five choices. Yet, the theorem amc12b_2020_p22 (Lean) asks to prove that for all values of the argument the function is smaller than the correct maximum. This is clearly insufficient without the prior knowledge of the correct answer (we can imagine that the solver could prove a weaker bound, but exceeded its timeout trying to prove correct bound).
   - (AIME problems are multiple-choice as well, but it is perhaps forgivable not to formalize them as such.)

Suggestion to the authors: Code such as Table 4 and the theorem on the circle and hyperbola (p.6) would be more readable with simple natural language annotations describing the meaning of each line, for the benefit of readers who are not familiar with all three systems or do not see the solution.

**Summary Of The Paper:**

This paper presents a new formal mathematics benchmark consisting of 488 statements expressed in three prominent theorem-proving/verification systems. Baseline ATP systems, notably GPT-f/PACT in Lean, are evaluated on this benchmark.

**Summary Of The Review:**

The main value of this work is in the set of formal Olympiad problem statements, which have not appeared in other datasets. There is little technical novelty in the ML algorithms and analysis of their performance, especially in comparison to [Han et al. 2021], on which this paper heavily relies.

---

> ### Author Response · Authors · 2021-11-20
> **Response to Reviewer Jr4E (1/2)**
>
> Thank you very much for your review and the time you dedicated to this.
>
> > The benchmark is not really as cross-system as claimed in the abstract.
>
> As we state in the “Porting effort” paragraph of Section 3, Isabelle is work-in-progress at the time of writing (validation set 35/244, test set 60/244), and we generally hope that MiniF2F will be a community-driven effort, in particular, to help span more systems, including HOL Light and Coq.
>
> We also would like to share that there is an active effort underway to port miniF2F to HOL Light that should be released before the end of the year.
>
> Please also note that the coverage is full for >1 systems (Metamath & Lean).
>
> > How were the Olympiad and Custom problems chosen?
>
> The Olympiad problems (AMC12 A/B, AIME, IMO) are chosen randomly while focusing on algebra, inequalities, and number theory problems.
>
> The CUSTOM problems are present mostly for historical reasons as they were problems we found interesting to tackle when we started evaluating our models. As an example, “Induction”, is a set of statements that can, but not necessarily, be solved by induction. All CUSTOM problems are drawn from high-school and undergraduate courses materials. We decided to keep these CUSTOM problems (despite them being a bit ad-hoc) as they span an interesting spectrum of exercises that we believe miniF2F benefits from.
>
> > The way in which multiple-choice problems are formalized gives additional information to the solver
>
> We added a paragraph “Formalization effort and challenges” to section 3 to better describe this challenge as well as others related to formalizing maths exercises, in particular:
>
> These multi-choice problems are generally straightforwardly formalizable by reformulating the statement using the right answer only and could be made “fair” in a competitive setup by formalizing all possible choices and running automated provers on all of them, attributing points only if proof of the correct answer is provided.
>
> > In Table 4, the AMC problem asks which value of an expression is possible (quantifier on a and b), but the formalization drops the quantifiers and asks to prove equality.
>
> The quantifiers are not dropped for amc12_2000_p11 (Table 4). The `(a b : R)` hypothesis is actually equivalent in Lean to a quantified version of the problem: in particular, one could transform the statement in its quantified version by the using `revert` tactic[0] and then recover the original statement using the `intros` tactic[1].
>
> [0] https://leanprover-community.github.io/mathlib_docs/init/meta/tactic.html#tactic.revert
>
> [1] https://leanprover-community.github.io/mathlib_docs/init/meta/tactic.html#tactic.intros
>
> > Problem 22 of AMC 12B 2020 asked to find the maximum value of a certain function, out of five choices. Yet, the theorem amc12b_2020_p22 (Lean) asks to prove that for all values of the argument the function is smaller than the correct maximum. This is clearly insufficient without the prior knowledge of the correct answer (we can imagine that the solver could prove a weaker bound, but exceeded its timeout trying to prove the correct bound).
>
> Agreed that for some problems, including ones that involve bounds, the solution we propose above for multi-choice questions in a competitive environment would not be complete. But the main purpose of miniF2F is to propose a benchmark to study neural theorem proving. Replicating the exact difficulty of the olympiads includes not being an explicit objective of the effort. We defer to the IMO Grand Challenge committee and other potential future committees to find appropriate solutions to these interesting and challenging issues.

---

> > ### Author Response · Authors · 2021-11-20
> > **Response to Reviewer Jr4E (2/2)**
> >
> > > Suggestion to the authors: Code such as Table 4 and the theorem on the circle and hyperbola (p.6) would be more readable with simple natural language annotations describing the meaning of each line, for the benefit of readers who are not familiar with all three systems or do not see the solution.
> >
> > Thank you for the useful suggestion. We included an informal version of the problem to Table 4, as well as what the example `algebra_sqineq_2unitcircatblt1` in section 4.3.1 represents in mathematics assertion.
> >
> > > The main value of this work is in the set of formal Olympiad problem statements, which have not appeared in other datasets. There is little technical novelty in the ML algorithms and analysis of their performance, especially in comparison to [Han et al. 2021], on which this paper heavily relies.
> >
> > We would like to stress that the main purpose of this paper is to present a benchmark for the community so that neural theorem proving in different systems can be studied comparatively. Studying the underlying systems that we present as baselines, including their performance in a more granular manner, is an explicit non-goal of this paper, especially since, as you point out, these systems were thoroughly analyzed in their respective presentations. The main goal of reporting numbers from these systems is to provide sensible baselines for researchers interested in neural theorem proving.

---

### Official Review · Reviewer_uyH2 · 2021-11-02

**Correctness:** 4
**Technical Novelty And Significance:** 4
**Empirical Novelty And Significance:** 3
**Recommendation:** 8
**Confidence:** 5

**Main Review:**

Strengths: (1) Since previous benchmarks of ATP mainly focus on basic math theorems, miniF2F fills the vacancy of the contest-level test suite for verifying the performance of theorem provers. I think this is an important step towards the goal of the grand-IMO challenge. (2) The cross-system design of miniF2F provides a good way to compare different formalizations and proving systems. (3) The experiment results demonstrate the importance of expert knowledge for theorem proving. Built with high-level tactics, GPT-f/Lean achieves better results than GPT-f/MM. The formal theorem provers also work better than the natural language-based problem solver.

Questions: (1) What are the meanings of "CUSTOM" and "Induction" in Table. 1. (2) What is the distribution of the number of theorems proved across different difficulty levels? (3) Personally, I am quite curious about your experience of formalizing these problems. What is the average time spent on one problem? Except for geometry and combinatorial problems, how large portion of problems could be formalized, and what would be the ultimate size of miniF2F in your expectation?





**Summary Of The Paper:**

This paper presents miniF2F, a test suite of Olympiad-level problems of theorem proving that is implemented in Metamath, Lean and Isabelle. MiniF2F contains 488 individual theorem statements that are formalized from Olympiad math contests. GPT-f models trained on Metamath and Lean are evaluated on this test suite.

**Summary Of The Review:**

Overall, I think miniF2F is an important benchmark that could help the community advance the research of theorem proving. I recommend accepting this paper.

===========================
I would like to maintain my old score of this paper after reading the authors' responses and other reviewers' comments.

---

> ### Author Response · Authors · 2021-11-20
> **Response to Reviewer uyH2**
>
> Thank you very much for your review and the time you dedicated to this.
>
>
> (1) The CUSTOM problems are present mostly for historical reasons as they were problems we found interesting to tackle when we started evaluating our models. As an example, “Induction”, is a set of statements that can, but not necessarily, be solved by induction. All CUSTOM problems are drawn from high-school and undergraduate courses materials. We decided to keep these CUSTOM problems (despite them being a bit ad-hoc) as they span an interesting spectrum of exercises that we believe miniF2F benefits from.
>
> (2) We added appendix B to this end to the updated version of the paper, including Table 5 and 6 reporting the number of proved statements from MATH Algebra/NumberTheory, split by difficulty.
>
> (3) On average, for trained undergraduate/graduate students as well as ourselves, we found that formalizing a statement roughly takes 15mn (we added a paragraph “Formalization effort and challenges” to that end in section 3). Reviewing a statement more like half of that. Not all statements are “directly” formalizable. Multi-choice problems are generally straightforwardly formalizable by using the right answer and could be made “fair” in a competitive setup by formalizing all possible choices and running the prover on all of them. Some problems are challenging to formalize because they rely on concepts not yet fully developed by the libraries associated with each formal system. Finally, some problems are not directly formalizable because they require human competitors to produce a witness or a set as an answer (e.g. find all … such that …). The best approximation we can do for these is to formalize the statement with the answer provided, turning such exercises into the generation of proof that the answer is correct, and if needed, that it is the unique one--which is, at times, a much easier exercise.
>
> Formal libraries are evolving rapidly and will likely soon cover all the materials needed to formalize any math competition up to the IMOs. Accepting the caveats associated with problems that require the explicitation of an answer, we believe pretty much any olympiad-type problem could be formalized, which represents at least 10k+ problems. All that being said, we believe the current size of MiniF2F is the minimal viable size to properly assess the capabilities of automated provers without being exposed to too much noise/variance, but ideally, we would love to see MiniF2F grow to cover a few thousand problems per test/valid set.

---

### Official Review · Reviewer_jbLg · 2021-11-04

**Correctness:** 4
**Technical Novelty And Significance:** 1
**Empirical Novelty And Significance:** 3
**Recommendation:** 6
**Confidence:** 3

**Main Review:**

Deep learning applied to theorem proving is I think one of its most exciting applications. The multiple different frameworks and datasets are a barrier to making progress in this area as a community and to that end this dataset is a significant step.

The methods the authors apply on the dataset are fairly state of the art and serve as a good baseline for someone wanting to make further progress. I do however think that some more analysis would be worthwhile. In particular, I think the authors should add the following

(1) Breakdown of the performance on the problems sourced from the MATH dataset by level of difficulty.
(2) A qualitative analysis of what kinds of problems the baseline models fail on and whether they fail on similar problems.

**Summary Of The Paper:**

The authors present miniF2F, a dataset of formalized mathematical problems drawn from diverse sources including IMO, AIME, AMC, undergraduate, and high school problems. The focus is on algebra, inequalities, and number theory as those problems are easier to formalize than for example, geometry or combinatorial problems. The formalization is done in Metamath, Lean, with efforts for Isabelle ongoing.

The authors run GPT-f on Metamath and Lean, and the tidy baseline (from the PACT paper) on the dataset and present results. They find that proving in Lean is vastly better for performance than Metamath which they conjecture is due to access to higher level tactics in Lean compared to Metamath.

**Summary Of The Review:**

Highly relevant new dataset with recent baselines run on it to get an idea of SOTA performance. However detailed analysis of the baselines on the dataset is lacking

---

> ### Author Response · Authors · 2021-11-20
> **Response to Reviewer jbLg**
>
> Thank you very much for your review and the time you dedicated to this.
>
> (1) We added appendix B to this end to the updated version of the paper, including Table 5 and 6 reporting the number of proved statements from MATH Algebra/NumberTheory, split by difficulty.
>
> (2) We also added a comment in appendix B providing a qualitative analysis for the failure modes of the baselines. Basically, problems requiring either multiple non trivial mathematical reasoning steps or the introduction of a challenging cut remain out of reach of the systems described here.

---

### Decision · Program_Chairs · 2022-01-20

**Decision:**

Accept (Poster)

**Comment:**

The paper presents miniF2F, a dataset of 488 highschool and college level math problems. The problems are fully formalized and include proofs in the Metamath, Lean and Isabelle theorem provers (as the reviewers pointed out, the support for Isabelle is limited, and that should be made clearer in the abstract). This multi-platform support is the main selling point of the benchmark, because it will make it possible to make direct comparisons among systems targeting different theorem provers.

The paper also does a good job discussing the benchmark selection and formalization process. This is important since some of the problems were translated from word problems.

As part of the rebuttal, the authors added extra information on the performance of the baselines and some qualitative details on how they fail.

Overall, there is agreement among the reviewers that this is a valuable benchmark that will enable comparisons among systems that today are very hard to compare.